# Drinking Molecular Hydrogen Water Is Beneficial to Cardiovascular Function in Diet-Induced Obesity Mice

**DOI:** 10.3390/biology10050364

**Published:** 2021-04-23

**Authors:** Haruchika Masuda, Atsuko Sato, Kumiko Miyata, Tomoko Shizuno, Akira Oyamada, Kazuo Ishiwata, Yoshihiro Nakagawa, Takayuki Asahara

**Affiliations:** 1Department of Physiology, Tokai University School of Medicine, 143 Shimokasuya, Isehara, Kanagawa 259-1193, Japan; asato@tsc.u-tokai.ac.jp (A.S.); miyata.kumiko@tsc.u-tokai.ac.jp (K.M.); shizuno@tsc.u-tokai.ac.jp (T.S.); ishiwata@jeff.co.jp (K.I.); 2Department of Innovative Medical Science, Tokai University School of Medicine, 143 Shimokasuya, Isehara, Kanagawa 259-1193, Japan; a.oya.lab@gmail.com (A.O.); asa777@is.icc.u-tokai.ac.jp (T.A.); 3Department of Opthalmology, Tokai University School of Medicine, 143 Shimokasuya, Isehara, Kanagawa 259-1193, Japan; nakayosi@is.icc.u-tokai.ac.jp

**Keywords:** obesity, molecular hydrogen, brown adipose tissue, white adipose tissue, cardiovascular disorders, metabolic syndrome

## Abstract

**Simple Summary:**

Molecular hydrogen (MH) reportedly exerts therapeutic effects against inflammatory diseases by alleviating oxidative stress. We investigated the cardiovascular protective effects of molecular hydrogen water (MHW) intake using high-fat diet-induced obesity (DIO) mice. We observed that MHW intake for 2 weeks did not improve the blood sugar level or body weight but decreased heart weight in DIO mice. Notably, MHW intake alleviated oxidative stress in both the heart and the adipose tissue. Moreover, it improved cardiac hypertrophy and restored left ventricular function in DIO mice, and promoted the histological conversion of energy storage to expenditure in adipose tissues with the upregulation of thermogenic and cardiovascular protective genes. Furthermore, MHW restored endothelial progenitor cell (EPC) bioactivity to maintain vascular homeostasis. Taken together, MHW intake exerts cardiovascular protective effects in DIO mice. Hence, MHW intake is a potential prophylactic strategy against cardiovascular disorders in metabolic syndrome.

**Abstract:**

Molecular hydrogen (MH) reportedly exerts therapeutic effects against inflammatory diseases as a suppressor of free radical chain reactions. Here, the cardiovascular protective effects of the intake of molecular hydrogen water (MHW) were investigated using high-fat diet-induced obesity (DIO) mice. MHW was prepared using supplier sticks and degassed water as control. MHW intake for 2 weeks did not improve blood sugar or body weight but decreased heart weight in DIO mice. Moreover, MHW intake improved cardiac hypertrophy, shortened the width of cardiomyocytes, dilated the capillaries and arterioles, activated myocardial eNOS-Ser-1177 phosphorylation, and restored left ventricular function in DIO mice. MHW intake promoted the histological conversion of hypertrophy to hyperplasia in white and brown adipose tissues (WAT and BAT) with the upregulation of thermogenic and cardiovascular protective genes in BAT (i.e., *Ucp-1*, *Vegf-a*, and *eNos*). Furthermore, the results of a colony formation assay of bone-marrow-derived endothelial progenitor cells (EPCs) indicated that MHW activated the expansion, differentiation, and mobilization of EPCs to maintain vascular homeostasis. These findings indicate that the intake of MHW exerts cardiovascular protective effects in DIO mice. Hence, drinking MHW is a potential prophylactic strategy against cardiovascular disorders in metabolic syndrome.

## 1. Introduction

Molecular hydrogen (MH), the lightest and most abundant chemical element, can act as an antioxidant by suppressing free radical chain reactions, thereby reducing the production of reactive oxygen species (ROS) [1,2,3,4]. The medical benefits of MH have prompted investigations on the therapeutic application of hydrogen gas against various inflammatory diseases [2].

The emerging evidence of MH as an antioxidant with the use of clinical and experimental animal models proves that administration of MH, either through hydrogen gas inhalation or drinking water, is a feasible strategy for the treatment of a variety of inflammatory diseases. For example, in animal models, MH gas inhalation and intake of MH water (MHW) were demonstrated to play a significant role in the prevention of various acute and chronic inflammatory diseases, including focal brain ischemia/reperfusion injury [1], myocardial ischemia [5], type 1 and 2 diabetes and insulin resistance [6,7], atherosclerosis [8], liver fibrosis [9], and hypertension [10]. Moreover, in a human study, a hydrogen-rich water bath once daily for 6 months was reported to decrease visceral fat with reductions in low-density lipoprotein and fasting blood glucose levels [11]. Clinically, MH intake was reported to be effective against metabolic syndrome (MetS) [12] and the adverse effects of chemotherapy [13] and radiotherapy [14].

Obesity-induced MetS is a well-recognized low-grade inflammatory disease from visceral adipose tissue (VAT) prevailing to systemic tissues with hyperglycemia and dyslipidemia, resulting in cardiovascular disease (CVD), diabetic cardiomyopathy, atherosclerosis, and ischemic diseases [15,16]. MHW intake reportedly alleviates glucose intolerance, dyslipidemia, and high serum levels of oxidized low-density lipoprotein via super oxide dismutase stimulation, leading to an antioxidant effect [12,17,18,19,20].

Moreover, in the animal experiments, the beneficial effects of MHW intake on MetS with type 2 diabetes mellitus in a mouse model of diet-induced obesity (DIO) or db/db mice genetically deficient in leptin receptor are reportedly induced by the upregulation of hepatocyte secretion of fibroblast growth factor-21 (FGF-21) [7,21] and enhanced expenditure of fatty acids and glucose [22]. The mechanism involved in the treatment is considered due to the early activation of a transcriptional coactivator, peroxisome proliferator-activated receptor (PPAR)-γ coactivator-1α (PGC-1α), subsequently triggering the PPARα pathway and resulting in FGF-21 upregulation [21,23]. Such a PGC-1α/PPARα/FGF-21 cascade enhances energy metabolism, with PGC-1α doing so through mitochondrial biogenesis regulation, leading to the improvement of fatty acid metabolism and glucose intolerance.

However, the effect of MH intake on adipose tissues and cardiovascular pathophysiology in MetS has yet to be revealed, albeit in terms of lipid and glucose metabolism. In addition, the effects of MHW intake against diabetic cardiomyopathy and vascular dysfunction in MetS remain to be investigated.

Human and mouse adipose tissues are heterogeneously classified as white adipose tissue (WAT) and brown adipose tissue (BAT). WAT is further classified as visceral and subcutaneous. Adipose tissues play opposite roles in energy metabolism: visceral WAT is involved in energy storage and glucose resistance, whereas subcutaneous WAT and BAT are associated with energy expenditure (thermogenesis) and glucose sensitivity [24,25]. Alternatively, WAT for energy storage, which consists of unilocular adipocytes (hypertrophy), can be converted to brown or beige WAT with multilocular adipocytes (hyperplasia), similar to BAT, for energy expenditure, thereby reducing the risks of MetS [26,27].

This study aimed to determine whether MHW intake is beneficial to the interplay between adipose tissue and the cardiovascular system in DIO mice as an animal model of MetS.

## 2. Materials and Methods

### 2.1. MHW and DIO Mice

The protocols of the animal experiments were approved by the ethics committee of Tokai University School of Medicine (protocol codes: 191091, 2020.3.30 and 201029, 2019.5.7) and were conducted in accordance with best practices for animal welfare, care, and accommodation (replacement, reduction, refinement). Male C57BL6/J mice were continuously fed a high-fat diet (5.25 Kcal/g; cat. no. D12492; Research Diets, Inc., New Brunswick, NJ, USA) from 4 weeks after birth as DIO mice at Charles River Laboratories Japan, Inc. (Kanagawa, Japan). DIO mice purchased at 10 weeks of age were acclimated for 2 weeks and were fed a high-fat diet ad libitum prior to experimentation. For the MHW or DGW preparation and supplementation (Figure 1a), MHW or DGW were prepared for 3 days and supplied to DIO mice for 3 to 4 days in one cycle. First, sterilized water was filled in a 1 L glass bottle with 3 MH supplier sticks (Dr. Suisosui, FDR Friendear Inc., Shinjuku, Tokyo, Japan), and retained at room temperature (RT) for 48 h (step 1). Next, MHW was transferred to a 0.2 L plastic bottle to degas H2 from MHW for DGW, and the glass bottle with the sticks was refilled with sterilized water for MHW (step 2). Each water type was retained at RT for 24 h. We measured the MH concentration and pH in each MHW water sample in a 1 L glass bottle (pre-MHW) or DGW in a 0.2 L plastic bottle (pre-DGW) at the end of step 2. Finally, MHW was transferred into a 0.5 L glass bottle with a MH supplier stick, and supplied to DIO mice for 72 h or 96 h as well as DGW in a 0.2 L plastic bottle (step 3). We also measured the MH concentration and pH in each water sample after supplementation to DIO mice (post-MHW or post-DGW). The cycles at steps 1–3 were sequentially repeated four times for 14 days (Figure 1b).

### 2.2. Hydrogen Concentration and pH of MHW

The dissolved hydrogen concentration in MHW was measured using a specific MH electrode, a DM-10B2 meter (ABLE Corporation, Tokyo, Japan). In brief, a 2-week supply of MHW and DGW was prepared every 3 to 4 days. When measuring the MH concentration in the MHW treated with MH supplier sticks and DGW, we first adjusted independently the standardized MH water in the absence of MH supplier sticks by bubbling 0.03 MPa of hydrogen gas into drinking water at 25 ℃ for 30 min. The MHW MH concentration before and after the addition bubbling of hydrogen gas was measured and standardized as 0% and 100%, respectively. Subsequently, the amounts of dissolved MH in the stored MHW and freshly prepared MHW with the sticks and DGW were calculated by measuring the MH percentages, based on the saturated MH concentration (1.56 ppm to 100% at 25 °C) in MHW at 25 °C. The pH was also measured using a pH meter (F-71SHoriba; Kyoto, Japan).

### 2.3. Blood Glucose and Body Weight (BW) Measurements

The right submandibular vein of non-fasting DIO mice with no diet limitation was punctured using a Goldenrod^TM^ animal lancet (point length, 5 mm; MEDIpoint Inc., Mineola, NY, USA) during the evening once per week, and the blood glucose level of oozing blood was measured using a Glutest sensor (Panasonic Healthcare Holdings Co., Ltd. Tokyo, Japan). BW was measured at the same time.

### 2.4. Measurement of Cardiac Function by Echocardiography

After 14 days of MHW intake, transthoracic echocardiography was performed using an Aloka IPC-1530 ultrasound monitor (Hitachi Healthcare, Tokyo, Japan) as previously reported [28]. The mice were lightly anesthetized (1.5% isoflurane) for the duration of the recordings. Heart rate was monitored simultaneously by electrocardiography.

Left ventricular (LV) end-diastolic diameter (LVEDD) and end-systolic diameter (LVESD) were used to calculate fractional shortening (FS) and the ejection fraction (EF) using the following formulas [28]:

FS(%) = [(LVEDD−LVESD)/LVEDD] × 100%,

EF(%) = [(LVEDD−LVESD)^3^]/LVEDD^3^] × 100%

Further, the LV mass index was calculated from the LV septal wall thickness (SWT) and LV posterior wall thickness (PWT) at end-diastole and then normalized by BW according to the following formula [28]:

LV mass (mg/g) = [(LVEDD + SWT + PWT)^3^ − LVEDD^3^] × 1.055/BW

Here, 1.055 (in mg/mm^3^) is the density of the myocardium. All echocardiography-derived values were obtained by averaging two or three measurements for each mouse.

### 2.5. Sample Preparation for Immunohistochemical Analysis, Quantitative Real-Time Polymerase Chain Reaction (qRT-PCR) and Endothelial Progenitor Cell (EPC) Colony Formation Assay

As individual experiments, the mice were anesthetized by intraperitoneal administration of pentobarbital sodium (60–70 mg/kg BW; Somnopentyl; Kyouritu Seiyaku Corporation, Tokyo, Japan), and blood was then aspirated by heart puncture using a 1 mL heparinized syringe. Immediately after excision, the BAT, WAT, and heart of each mouse were weighed and each of the BAT or WAT was divided for immunohistochemical analysis and qRT-PCR. As a separate experiment, the aspirated blood and the tibia and femur, excised from left side limb, were applied for endothelial progenitor cell (EPC) colony formation assay.

### 2.6. Immunohistochemical Analysis

After fixation with 4% paraformaldehyde overnight, the BAT, WAT, and heart were collected, embedded in paraffin, sectioned, and stained with hematoxylin and eosin (H&E) for histologic analysis. Deparaffinized tissue samples were blocked with avidin and biotin solution for 15 min using an Avidin/Biotin Blocking Kit (SP-2001; Vector Laboratories, Burlingame, CA, USA) as described in the supplemental protocol. Subsequently, the samples were incubated with isolectin GS1-B4 biotin conjugate (Vector Laboratories, Burlingame, CA, USA) diluted (1:100) with PBST (0.2% Tween 20, 0.1% CaCl_2_, and 0.1% MgCl_2_ in PBS) at RT for 2 h, washed, and then incubated with streptavidin-Alexa Fluor 488 diluted (1:500) with PBST at RT for 1 h. The stained samples were mounted with glycerol mounting medium with 1, 4-Diazabicyclo[2.2.2] octane (DABCO )(Sigma-Aldrich, St. Louis, MO, USA). Separately, the deparaffinized samples were also stained with anti-alpha smooth muscle actin-Cy3 conjugate (Cy3-αSMA; Sigma-Aldrich), blocked with 10% normal goat serum in PBST without Ca^2+^ or Mg^2+^ at RT for 30 min, incubated with Cy3-αSMA diluted (1:200) with 5% normal goat serum in PBST without Ca^2+^ or Mg^2+^ at RT for 2 h, washed with PBS, and mounted with glycerol mounting medium with DABCO^TM^. Independently, the deparaffinized samples were stained with anti-8-hydroxy-2′-deoxyguanosine (8-OHdG) monoclonal antibody (Japan Institute for the Control of Aging, Shizuoka, Japan) following the same blocking protocol. The samples were incubated with anti-8-OHdG antibody 5 μg/mL with 5% normal goat serum in PBST without Ca^2+^ or Mg^2+^ at 4 °C overnight, washed with PBS, stained with DAKO EnVision™+ System, Peroxidase (DAKO EnVision™+ System; HRP, DAKO Japan, Tokyo, Japan), and DAB (3,3′-Diaminobenzidine). After sequential H-staining, the samples were mounted with glycerol mounting medium with DABCO. The stained tissue samples were observed under a fluorescence microscope (FSX-100; Olympus Corporation, Shinjuku, Tokyo, Japan), and analyzed using the cellSens software (Olympus Corporation, Shinjuku, Tokyo, Japan).

### 2.7. qRT-PCR

Parts of the BAT and WAT were incubated in 1 mL of RNAlater™ Stabilization Solution (Thermo Fisher Scientific, Waltham, MA, USA) at 4 °C overnight, and then stored at −80 °C until use. Total RNA was isolated using QIAzol Lysis Reagent (QIAGEN, Hilden, Germany). Contaminated genomic DNA was digested with DNase I (Thermo Fisher Scientific, Waltham, MA, USA) at 37 °C for 15 min. DNase-I-treated total RNA was purified by phenol extraction and ethanol precipitation. Then, 2 µg of purified total RNA was reverse-transcribed into complementary DNA (cDNA) with the Applied Biosystems™ High-Capacity cDNA Reverse Transcription Kit (Thermo Fisher Scientific). The cDNA mixture was diluted 10-fold after first-strand cDNA synthesis and then amplified by qRT-PCR using the ABI PRISM^®^ 7700 Sequence Detection System with TaqMan probes (Thermo Fisher Scientific) according to the manufacturer’s protocol. Relative mRNA levels were calculated using the ∆∆Ct method and normalized against mouse 18S rRNA. All primers and probes are listed in Appendix A.

### 2.8. Isolation of Peripheral Blood Mononuclear Cells (PBMCs) and Bone Marrow (BM)-Derived c-Kit + /Sca-1 + Lineage-Negative Cells (BM-KSL Cells)

The aspirated blood was collected into a 1.5 mL tube per mouse. Then, the femur and tibia from the left limb of each mouse were dissected and transferred to a 15 mL conical tube containing Hanks’ balanced salt solution supplemented with 1% fetal bovine serum (FBS). PBMCs and BM-KSL cells, which are stem cell populations of EPCs, were isolated as previously reported [29].

Lineage-negative BM cells (BM-Lin-) were incubated with saturating concentrations of directly labeled anti-c-Kit (dilution, 1:25; BD Biosciences, Franklin Lakes, NJ, USA) and anti-Sca-1 antibodies (at 1:25 dilution) (BD Biosciences) for 30 min on ice. Afterward, BM-KSL cells were isolated by live sterile cell sorting using a BD FACSAria™ Fusion flow cytometer (Becton Dickinson, Franklin Lakes, NJ, USA).

### 2.9. EPC Colony Formation Assay

As previously described [29], PBMCs or isolated BM-KSL cells were cultured in medium containing methylcellulose (MethoCult™ SF M3236; StemCell Technologies, Vancouver, BC, Canada) with 20 ng/mL of stem cell factor (Peprotech Inc., Rocky Hill, NJ, USA), 50 ng/mL of vascular endothelial growth factor (Peprotech Inc.), 20 ng/mL of interleukin-3 (Peprotech Inc.), 50 ng/mL of basic fibroblast growth factor (Peprotech Inc.), 50 ng/mL of epidermal growth factor receptor (Peprotech Inc.), 50 ng/mL of insulin-like growth factor-1 (Peprotech Inc.), 2 U/mL of heparin (Ajinomoto Co., Inc., Tokyo, Japan), and 10% FBS on a 35-mm Falcon™ Primaria™ Cell Culture Dish (Corning Incorporated, corning, NY, USA) for 8 days. PBMCs were seeded at 5 × 10^5^ cells per dish and BM-KSL cells at 500 cells per dish. The colony-forming units (CFUs) of EPCs were identified as small-EPC-CFUs (primitive EPC-CFUs; pEPC-CFUs) or large-EPC-CFUs (definitive EPC-CFUs; dEPC-CFUs) by visual inspection with an inverted microscope at 40× magnification as previously reported [29]. The differentiation ratio was calculated as the percentage of dEPC-CFUs of the total EPC-CFUs.

### 2.10. Statistical Analysis

Statistical analysis was conducted using the Prism 9.0 software (GraphPad Software, Inc., San Diego, CA, USA). Two-way analysis of variance with Holm–Šídák’s multiple comparisons test was used to identify differences between the DGW and MHW groups at each time point or between or among the time points (Figure 1c,d, Figure 2). The Mann–Whitney U test was used to compare the values between two groups. A probability (*p*) value of <0.05 was considered statistically significant. Data are expressed as mean ± standard deviation (SD).

## 3. Results

### 3.1. MH Concentration in Drinking Water

The dissolved MH concentrations in freshly prepared MHW and DGW were measured prior to availability to the DIO mice every 3 to 4 days for the entire 2-week observation period (Figure 1a,b). The dissolved MH concentration in freshly prepared MHW in a 1 L glass bottle with three MH-producing sticks was 0.88 ± 0.25 ppm. Contrarily, MHW was significantly degassed to 0.01 ± 0.01 ppm by transferring into a 200 mL plastic bottle for 24 h (pre-DGW). Moreover, the MH concentration in the DGW was not detected after supplementation (post-DGW). Even after supplying to mice for 3 to 4 days, the dissolved MH concentration in MHW was maintained at 0.61 ± 0.26 ppm in a 0.5 L glass bottle with one stick (Figure 1c). However, no significant change was observed in the pH of the MHW before or after supplementation. In addition, there was no significant difference in the pH between the MHW and DGW before and after supplementation. These findings indicate that the MH concentration, rather than the pH, was the effector in this study (Figure 1d, Appendix A).

### 3.2. MHW Had no Effect on BW or Postprandial Blood Sugar (PPBS) of DIO Mice

The BW of mice in the MHW and DGW groups significantly increased on days 7 and 14 vs. day 0 (before drinking MHW or DGW), although no significant difference was observed between groups at any time point (Appendix A). However, the postprandial blood sugar (PPBS) of DIO mice in the MHW group decreased on day 14 vs. day 0, whereas there was no significant difference in BW and PPBS between groups on any day. These findings indicate that MHW had no significant effect on BW or PPBS during the observation period, although PPBS was often lower in the MHW group compared with the DGW group (Figure 2, Appendix A).

### 3.3. MHW Morphometrically Changed the Anti-Metabolic Phenotype of Adipose Tissue in DIO Mice

After continuous MHW supplementation for 2 weeks, there was no change in the WAT mass of DIO mice (Figure 3a,b), although the BAT mass increased by 1.3-fold (Figure 3c,d). The percentage of WAT weight per kg BW was 8.32% ± 0.52% in the DGW group and 8.00% ± 0.62% in the MHW group, whereas the percentage of BAT was 0.40% ± 0.05% in the DGW group and 0.53% ± 0.07% in the MHW group. The circumferential lengths of WAT adipocytes significantly shortened in the MHW group compared with the DGW group (221.9 ± 33.5 vs. 294.7 ± 46.5 μm, respectively) (Figure 3e,f), indicating that MHW intake was associated with a decrease in the size of WAT adipocytes in DIO mice. In other words, MHW intake was associated with the transition from unilocular (hypertrophic) to multilocular (hyperplasia) WAT adipocytes in DIO mice. Similarly, it changed the phenotype of BAT adipocytes from hypertrophic to hyperplastic (Figure 3g). Notably, 8-OHdG+ cells indicating ROS-induced DNA damage decreased (17.4 ± 6.6/mm^2^ vs. 43.5 ± 10.5/mm^2^) in the MHW group compared with that in the DGW group (Figure 3h,i, Appendix A), whereas 8-OHdG- cells without the damage increased (11.6 ± 3.4/mm^2^ vs. 3.75 ± 2.5/mm^2^). These findings indicate that MHW intake alleviated the oxidative stress in WAT in DIO mice.

### 3.4. MHW Intake Upregulated the Expression of Thermogenic- and Angiogenic-Related Genes in the BAT of DIO Mice

MHW intake was associated with 1.98-fold increase in *eNos* expression in the WAT of DIO mice. Conversely, it significantly increased the mRNA levels of thermogenic-related *(eNos, Ucp-1*) and angiogenic-related (*eNos, Vegf*) genes in the BAT of DIO mice compared with DGW by 1.63-, 1.64-, and 1.57-fold for *Ucp-1*, *eNos*, and *Vegf*, respectively (Figure 4). MHW intake tended to upregulate the expression of the thermogenic gene *iNos* and downregulate that of the inflammatory cytokine genes *Tnf-**α* and *Il-1β* in both the WAT and BAT, although there were no significant changes (Appendix A, Appendix A). These findings indicate that MHW intake accelerates thermogenesis and angiogenesis, resulting in the anticipated energy expenditure as well as NO production in adipose tissues.

### 3.5. MHW Intake Alleviated Metabolic Cardiomyopathy in DIO Mice

MHW intake contributed to the morphometric and functional improvements in metabolic cardiomyopathy in DIO mice.

(1)Morphometric improvement in cardiac hypertrophy

Assessing H&E-stained cardiomyocytes at a magnification of 40×, the average width of cardiomyocytes in the MHW group was significantly narrowed by 0.78-fold compared with that in the DGW group (11.8 ± 0.58 vs. 15.1 ± 0.69 µm, respectively) (Figure 5a,b).

(2)MHW intake promoted eNOS phosphorylation in cardiomyocytes of DIO mice

The LV area occupied by cardiomyocytes with phosphorylated Ser1177 of eNOS was significantly extended by 1.7-fold in the MHW group compared with the DGW group (50.5% ± 5.1% vs. 29.2% ± 5.1% per section, respectively) (Figure 5c,d).

(3)Cardiac arteriolar and capillary dilatation in DIO mice in the MHW group

No significant difference was observed in the counts of cardiac capillaries stained with isolectin B4-Alexa Fluor 488 between the MHW and DGW groups (220.7 ± 60.2 vs. 215.3 ± 54.6/mm^2^, respectively) (Figure 5e,f). Conversely, the average short-axis diameter was significantly increased by 1.3-fold in the MHW group compared with the DGW group (4.2 ± 0.52 vs. 3.3 ± 0.65 µm, respectively) (Figure 5e,g).

Further, the average short-axis diameter and count of arterioles stained with α-SMA antibody were calculated. These features of the cardiac arterioles were similar to those of the capillaries. No significant difference was observed in the arteriolar counts between the MHW and DGW groups (52.0 ± 14.0 vs. 54.8 ± 7.6/mm^2^, respectively) (Figure 5h,i). Conversely, the average short-axis diameter was significantly increased by 1.4-fold in the MHW group compared with that in the DGW group (8.0 ± 1.67 vs. 5.6 ± 0.92 µm, respectively) (Figure 5h,j).

(4)MHW intake alleviated cardiac oxidative stress in DIO mice in the MHW group

Similar to the features observed in the case of WAT, 8-OHdG+ cells tended to decrease in the MHW group compared with the DGW group (1977 ± 361/mm^2^ vs. 3021 ± 1280/mm^2^), although the difference was not statistically significant (Figure 5k,l, Appendix A). Notably, 8-OHdG- cells increased in the MHW group compared with the DGW group (620 ± 307/mm^2^ vs. 124 ± 119/mm^2^), indicating that MHW intake alleviated the cardiac oxidative stress in DIO mice.

(5)MHW intake improved cardiac function in DIO mice

Cardiac function was assessed by echocardiographic measurement of the LV parameters, namely, SWT, LVEDD, LVESD, and PWT, in M-mode (Figure 6a,b). There was no significant difference in the heart rate between the two groups (Figure 6c, Appendix A). Notably, the LV mass per g BW was decreased by 0.8-fold in the MHW group compared with that in the DGW group (3.64% ± 0.54% vs. 4.57% ± 0.20%, respectively) (Figure 6d, Appendix A). The EF% significantly recovered in the MHW group compared with the DGW group (40.1% ± 5.54% vs. 22.8% ± 2.53%, respectively) (Figure 6e, Appendix A), as did EF% (90.3% ± 4.1% vs. 75.3% ± 4.4% respectively) (Figure 6f).

### 3.6. EPC Bioactivity Improved in the MHW Group

pEPC-CFUs, dEPC-CFUs, and total EPC-CFUs were increased by 2.0-, 4.0-, and 2.5-fold per mL of circulating blood, respectively, in the MHW group compared with those in the DGW group (Figure 7a,b, Appendix A). The generation of dEPC-CFUs and total EPC-CFUs by 500 BM-KSL cells was 1.4- and 1.1-fold greater in the MHW group compared with the DGW group, whereas no significant difference was observed in the number of pEPC-CFUs (Figure 7c, Appendix A). Further, the differentiation grade of EPCs from BM-KSL cells was calculated as the percentage of dEPC-CFUs of the total EPC-CFUs generated from 500 BM-KSL cells. The differentiation grade of EPCs was significantly higher in the MHW group than in the DGW group (27% vs. 22%, respectively) (Figure 7d, Appendix A). pEPC-CFUs, dEPC-CFUs, and total EPC-CFUs from total KSL cells isolated from the right femur and tibia were increased by 1.3-, 1.8-, and 1.4-fold in the MHW group compared with the DGW group (Figure 7a,e, Appendix A). These findings indicate that MHW intake restored EPC mobilization into the circulation as well as expansion and differentiation from BM-derived vasculogenic stem cells.

## 4. Discussion

This study aimed to determine whether MHW intake can alleviate cardiovascular dysfunction in obesity associated with MetS in DIO mice. Interestingly, the results revealed that even a relatively short term (2 weeks) MHW intake restored cardiac function and vascular bioactivity in DIO mice, linking the anti-oxidative and -inflammatory effects to the adipose tissues. MHW intake by DIO mice with chronic cardiovascular disorders increased eNOS phosphorylation in cardiomyocytes, which led to a significant cardiovascular dilatation of capillaries and arterioles and subsequent cardioprotection. Further, MHW intake restored the kinetics of EPCs (i.e., expansion, differentiation, and mobilization from the BM). The molecular mechanism of such cardiovascular protective effects by MHW has been associated with adipose tissue remodeling but not alleviation of hyperglycemia.

### 4.1. BAT Activation and WAT Browning

As reported in some previous studies, the BAT or WAT of DIO mice behaves as a critical source of inflammation, giving rise to diabetic cardiomyopathy accompanied by obesity [30,31,32]. MHW intake has been reported to alleviate inflammation by reducing the protein levels of TNF-α and IL-1β in the rat injured lung through the inhibition of NF-kβ [33].

BW was similarly increased in both the DGW and MHW groups for the entire 2-week observation period, with % ratio of BAT mass to BW significantly increased in the MHW group and no change in the % ratio of WAT mass (Figure 2a, Figure 3).

While there was no change in PPBS in the DGW group over the 2-week observation period, this parameter decreased in the MHW group at day 14 compared with pretreatment at day 0 (Figure 2b).

In addition, MHW intake for 14 days did not significantly decrease the expression levels of the proinflammatory cytokine genes *Tnf-**α* and *Il-1β* in the BAT or WAT compared with the DGW group, although the gene expression levels tended to decrease, which may have become apparent due to the longer treatment period.

Kamimura et al. [21] reported that compared with degassed water, MHW intake even for 74 weeks had no effect on the BW of DIO mice. Moreover, the MHW intake improved hyperglycemia, hyperinsulinemia, and the plasma triglyceride level with decreasing BW in diabetic db/db mice with genetically engineered leptin receptor deficiency at 3 months [7]. Taking these references into account, in the present study, a more prolonged MHW intake period might be necessary to improve hyperglycemia levels.

Additionally, MHW intake was reported to upregulate the gene expression of ghrelin [34], which is an orexigenic (appetite-stimulating) and adipogenic peptide hormone secreted in the stomach. Moreover, a high-fat diet increases ghrelin-expressing cells in the gastric mucosa of mice, which contributes to obesity [35]. As noted in previous studies, the possible increase in ghrelin by MHW intake and feeding of a high-fat diet might extinguish the beneficial effect of MHW on BW or WAT mass.

Conversely, the expression of genes (*Pgc-1a, Fgf21*) associated with lipid metabolism for energy expenditure in the liver of db/db mice was definitively promoted by day 14 after the initiation of MHW intake [21]. Alternatively, MHW intake restored the excess amount of lipids stored in the liver of DIO mice by day 14 [7]. Collectively, these findings and the results of the present study indicate that MHW intake could restore lipid metabolism toward energy expenditure more favorably than glucose metabolism.

Notably, MHW intake improved the ROS-induced DNA damage in WAT in DIO mice, indicating that MHW exerted an antioxidative effect (Figure 3h,i). MHW intake was also associated with upregulated expression of the thermogenic genes *Ucp-1*, *Vegf*, and *eNos*, which promoted adipose tissue remodeling and subsequent BAT activation or WAT browning in DIO mice [36,37,38].

Further, in the present study, *eNos* expression was increased, which promoted the expression of *Ucp-1* and *Vegf* in the WAT of DIO mice in the MHW group (Figure 4), indicating that MHW intake accelerated WAT browning and BAT activation to evade CVD in DIO mice [39,40,41].

### 4.2. Cardiovascular Protective Effects

Emerging evidence proves that BAT activation or WAT browning in obesity is beneficial to cardiovascular health and decreases the risk of CVD [16,42].

With regard to the underlying molecular mechanisms, Thoonen et al. [41] reported that UCP-1 in BAT can reverse catecholamine-induced cardiomyopathy. BAT transplantation isolated from wild-type mice recovers catecholamine-induced cardiomyopathy, which leads to fibrosis in *Ucp-1-/-* mice. Moreover, overexpression of *Ucp-1*, which is a specific protein uncoupler of oxidative phosphorylation capable of collapsing the proton electrochemical gradient, also prevented the effects of hyperglycemia [43]. In addition, ablation of *Ucp-1* expression in BAT promoted ROS generation [44].

More recently, WAT (cardiac VAT) in obese rats with disturbed *Ucp-1* gene expression was reported to activate hypertrophic cardiomyocytes with impaired viability and fibrosis in a myocardial organ culture system [45].

Considering the previous studies, the upregulation of *Ucp-1* expression in BAT and WAT browning of DIO mice via MHW intake is considered to induce protective effects against diabetic cardiomyopathy via the upregulation of *Ucp-1* expression in BAT, thereby probably decreasing ROS generation and activating eNOS in BAT (Figure 4). As a result, this mechanism conceivably repairs cardiac function in DIO mice treated with MHW. Of note, in the present study, MHW intake promoted eNOS activation in the myocardium (eNOS-Ser 1177 phosphorylation) (Figure 5c,d) and adipose tissue, especially BAT (*eNos* expression) (Figure 4) in DIO mice. However, future studies are required to determine whether MHW can boost eNOS phosphorylation in the myocardium directly or through *Ucp-1* upregulation in BAT or WAT browning.

The activation (phosphorylation) of eNOS-Ser 1177 in cardiomyocytes in DIO mice in the MHW group (Figure 5e–j) promoted NO generation and inversely inhibited ROS production (Figure 5k,l), which enhanced NO bioactivity to improve cardiac function with vascular dilation as cardiovascular protective effects [46,47,48]. In fact, echocardiography revealed LV systolic function in DIO mice in the MHW group with a decrease in LV mass, indicating recovery from cardiac hypertrophy (Figure 6).

### 4.3. Activation of EPC Bioactivity

In the present study, MHW intake promoted the mobilization of EPCs into the circulation, as well as expansion and differentiation in BM (Figure 7).

Previous studies have reported the dysfunction of BM-derived EPCs in MetS in both humans and mice [16,49,50].

In the present study, MHW intake promoted eNOS activation in the adipose tissues and myocardium in DIO mice, indicating NO generation. Accordingly, exogenous NO generation may restore EPC dysfunction [51]. Further, the antioxidant effect of MHW in DIO mice might induce endogenous NO production via eNOS activation of EPCs, thereby autonomously improving the bioactivities of EPCs.

### 4.4. The Unlikely Alkaline Effect of MHW

The MHW utilized in this study also contained Mg(OH)_2_ produced by the reaction of H_2_O with Mg in the MH supplier sticks, leading to an increase in pH to around 10, thereby producing an alkaline solution. Hence, future studies are required to investigate the synergistic effects of hydrogen molecules and alkaline ions in MHW. In addition, Jackson et al. [52] recently reported that hydrogen-rich water improved fatty liver in a high-fat diet-induced model of nonalcoholic liver disease but not alkaline-electrolyzed water at pH 11, which was likely due to H_2_ in hydrogen-rich water. Previous studies demonstrated, as the beneficial mechanism, that H_2_, but not electrolyzed water, upregulated peroxisomal fatty acyl-CoA oxidase, a rate-limiting enzyme for beta-oxidation of very long chain fatty acids in peroxisomes, contributing to lipid metabolism.

Of note is that, in the present study, no difference was observed in the pH values of DGW and MHW at both pre- and post-supplementation. Accordingly, those findings are concluded to have originated via MHW intake.

### 4.5. Study Limitations

The possibility of a synergistic effect of MHW with alkaline Mg(OH)_2_ could not be excluded, as the effect of DGW was not compared with that of untreated water. In addition, with regard to the mechanism of MHW, the effect of UCP-1 in adipose tissues on EPC bioactivity in DIO mice remains unknown.

## 5. Conclusions

For the first time, we show that MHW intake by DIO mice exerted cardiovascular protective effect via WAT browning and BAT activation with energy expenditure phenotypes. Consequently, MHW provides a prophylactic strategy against CVD in MetS patients. 

## Figures and Tables

**Figure 1 biology-10-00364-f001:**
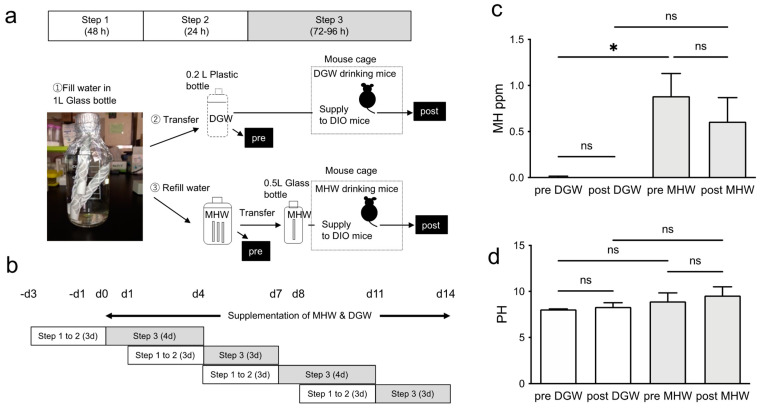
Preparation and supplementation of MHW. (**a**) MHW preparation and supplementation protocol with three MH supplier sticks (Dr Suisosui, FDR Friendear Inc., Shinjuku, Tokyo, Japan) and DGW for one cycle (**a**), and for the repeated cycles for 14 days (**b**). (**c**) MH concentration (ppm) at pre- and post-supplementation of MHW. *n* = 4 in each of the pre or post steps; * *p* < 0.05; ns, not significant. (**d**) pH levels at pre- and post-supplementation of MHW and DGW. *n* = 4 in each of the pre or post steps; ns, not significant.

**Figure 2 biology-10-00364-f002:**
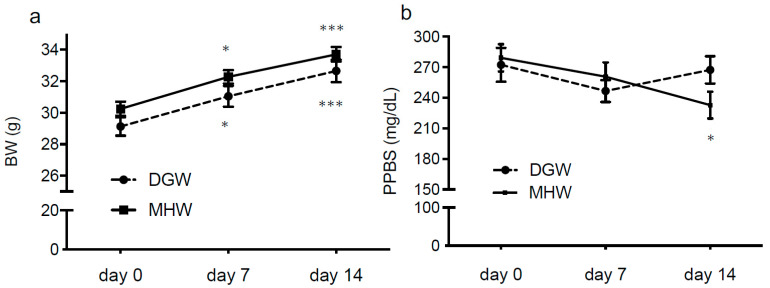
The transition of BW and PPBS in DIO mice treated with MHW vs. DGW for 2 weeks. BW (**a**) and PPBS (**b**) in DIO mice treated with MHW or DGW for 14 days. *n* = 12; * *p* < 0.05, *** *p* < 0.001 vs. day 0 in each group.

**Figure 3 biology-10-00364-f003:**
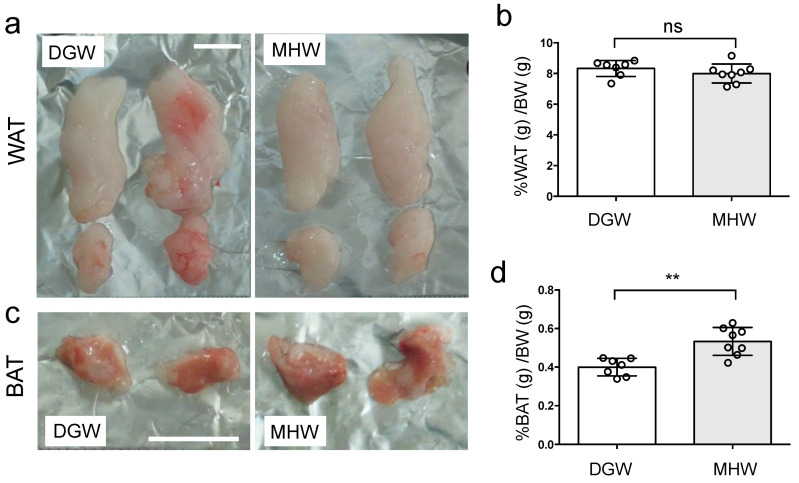
The features of WAT and BAT in DIO mice after treatment with MHW vs. DGW for 2 weeks. The representative features of adipose tissue masses of WAT (**a**) and BAT (**c**) in DIO mice treated with MHW or DGW. The percentages of WAT (**b**) or BAT (**d**) per kg BW in DIO mice treated with either MHW or DGW. *n* = 7 or 8; ** *p* < 0.01; ns, not significant. (**e**) Representative H&E-stained images of WAT adipocytes in DIO mice treated with either MHW or DGW. (**f**) The average circumferential length of WAT adipocytes in the MHW and DGW groups. *n* = 6; *** *p* < 0.001. (**g**) Representative H&E-stained images of BAT in the DGW and MHW groups; ns, not significant; *n* = 6; *** *p* < 0.001. (**h**) Hematoxylin and 8-OHdG-stained features of WAT in MHW- or DGW-treated DIO mice. The image surrounded with a yellow line in the DGW or MHW panels is magnified in each image in the yellow-framed insert at the bottom left corner. The double-stained (dark brown) and hematoxylin-stained (blue) nuclei indicate the 8-OHdG+ and 8-OHdG- cells. (**i**) 8-OHdG+ and 8-OHdG- cell densities in WAT of the MHW and DGW groups; *n* = 4; * *p* < 0.05.

**Figure 4 biology-10-00364-f004:**
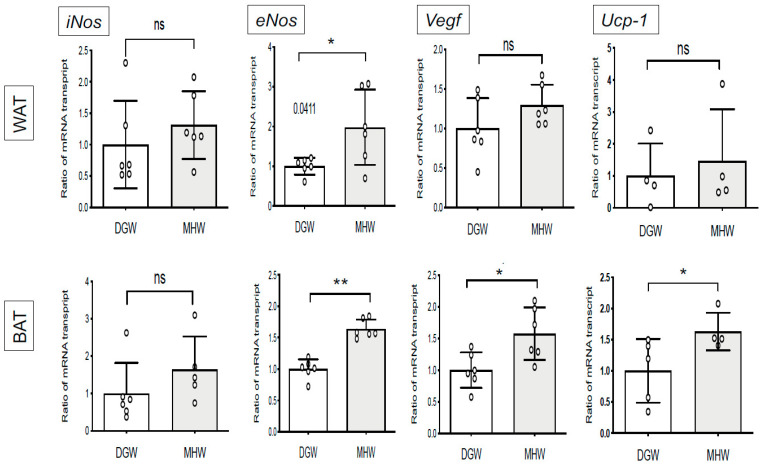
The genes expressed in BAT of DIO mice drinking MHW vs DGW for 2 weeks. Y-axis indicates the transcriptional ratio of expressing genes (*eNos*, *Vegf*, *or Ucp-1*) in WAT or BAT of DIO mice drinking MHW compared with DGW. The bar graphs show mean ± SD. * *p* < 0.05, ** *p* < 0.01; ns, not significant.

**Figure 5 biology-10-00364-f005:**
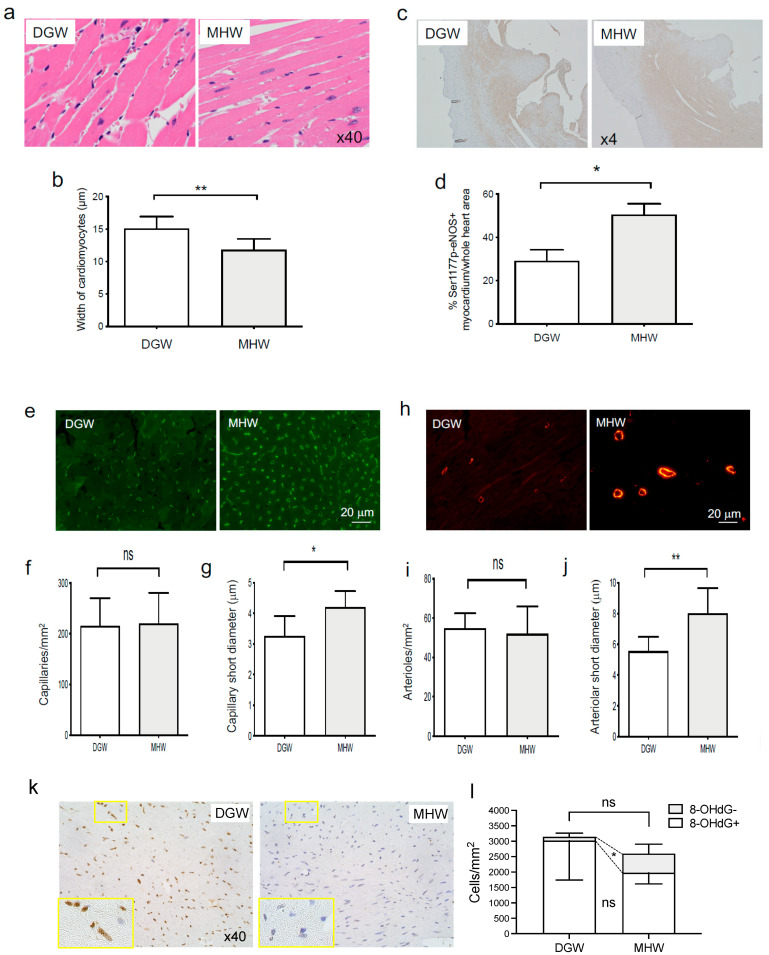
The morphological features of the hearts of DIO mice drinking MHW for 14 days. (**a**,**b**) H&E staining features and widths of the cardiomyocytes of DIO mice in the DGW or MHW group. The bar graph indicates the width of cardiomyocytes in each image. *n* = 7 or 8; ** *p* < 0.01. (**c**,**d**) Cross-sectional H&E-stained images of LV Ser1177p-eNOS^+^ myocardium. The bar graph indicates the percentage of the Ser1177p-eNOS^+^ myocardium area to the cardiac area in each image. *n* = 4 or 5; * *p* < 0.05. (**e**–**g**) Cross-sectional images of LV capillaries stained with isolectin B4-Alexa Fluor 488. The bar graphs indicate capillary counts/mm^2^ and intraluminal short-axis diameters. (**e**–**g**) Cross-sectional images featuring LV capillaries stained with isolectin B4-Alexa Fluor 488. The bar graphs indicate capillary counts/mm^2^ and intraluminal short-axis diameters. *n* = 6 or 7; ns, not significant; * *p* < 0.05. (**h**–**j**) Cross-sectional images featuring LV arterioles stained with αSMA-Alexa Fluor 594. The bar graphs indicate arteriolar counts/mm^2^ and intraluminal short-axis diameters. *n* = 7 or 8; ns, not significant; ** *p* < 0.01. (**k**) Hematoxylin and 8-OHdG-stained features of the heart in MHW- or DGW-treated DIO mice. The yellow-framed image in DGW or MHW panels is magnified in each image in the yellow-framed insert at the bottom left corner. The double-stained (dark brown) and hematoxylin-stained (blue) nuclei indicate the 8-OHdG+ and 8-OHdG- cells. (**l**) 8-OHdG+ and 8-OHdG- cell densities in the heart of mice in the MHW and DGW groups; *n* = 4; * *p* < 0.05.

**Figure 6 biology-10-00364-f006:**
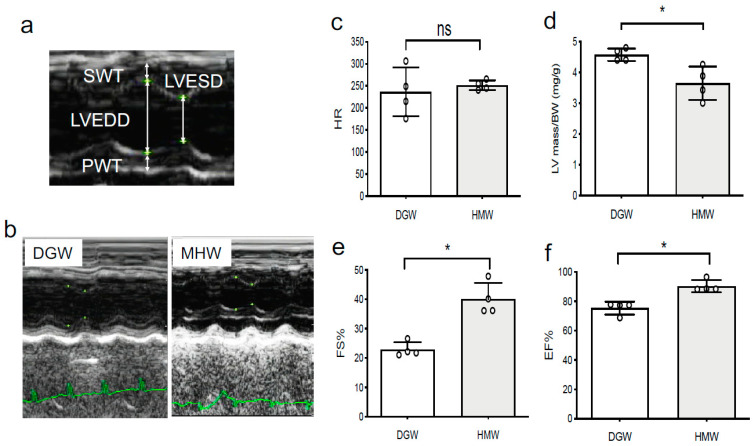
The physiological cardiac function in DIO mice treated with DGW or MHW for 14 days. (**a**) Parameters (LVEDD, LVESD, LV SWT, and LV PWT) to evaluate cardiac function in the M-mode view of mouse echocardiography. (**b**) Cardiac electrocardiography M-mode views of DIO mice in the DGW and MHW groups. (**c**) Heart rate. (**d**) LV mass/BW (mg/g) calculated using the formula LV mass (mg/g) = [(LVEDD + SWT + PWT)^3^ − LVEDD^3^] × 1.055/BW. (**e**) % fractional shortening (FS%) = [(LVEDD − LVESD)/LVEDD] × 100%. (**f**) %ejection fraction (EF%) = [(LVEDD − LVESD)^3^]/LVEDD^3^] × 100%; ns, not significant; * *p* < 0.05.

**Figure 7 biology-10-00364-f007:**
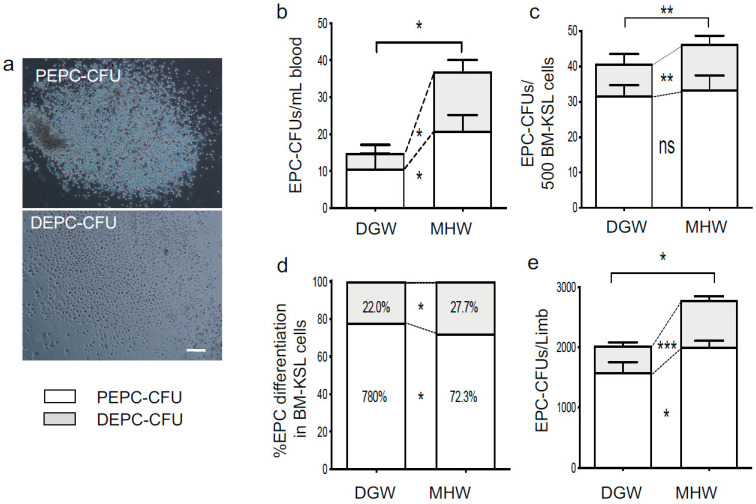
The biological properties of EPCs in DIO mice treated with DGW or MHW for 14 days. (**a**) Representative images of pEPC-CFUs and dEPC-CFUs in methylcellulose culture medium. (**b**–**e**) The counts and differentiation degree of EPC-CFUs between the DGW and MHW groups. EPC-CFU counts per mL blood (**b**), originated from 500 BM-KSL cells (**c**), originated from whole BM-KSL cells in the left femur and tibia (**e**). (**d**) The EPC differentiation degree calculated using the ratio of dEPC-CFUs/total EPC-CFUs (pEPC-CFUs + dEPC-CFUs) in (**c**). The clear and gray columns indicate pEPC-CFUs and dEPC-CFUs, respectively; ns, not significant; * *p* < 0.05, ** *p* < 0.01, *** *p* < 0.001.

## Data Availability

The data presented in this study are available on reasonable request from the corresponding author.

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
