# Peer review of "Drinking Molecular Hydrogen Water Is Beneficial to Cardiovascular Function in Diet-Induced Obesity Mice"

_biology, 2021, doi:10.3390/biology10050364_

Round 1
Reviewer 1 Report
The manuscript entitled “Drinking Molecular Hydrogen Water is Beneficial to Cardio vascular Function in Diet-Induced Obesit” by Haruchika Masuda et al, shows that molecular hydrogen water (MHW) is a potential prophylactic strategy against cardiovascular disorders in metabolic syndrome. The purpose of this study is to determine whether MHW intake is beneficial to the interplay between adipose tissue and the cardiovascular system in Diet-Induced Obesit (DIO) mice as an animal model of metabolic syndrome (MetS),.
It is interesting and valuable results, however, there are some comments and question as below.
Major points:
- In Abstract and Introduction: Now there are arguments against the description on “a scavenger of reactive oxygen species” as a effect of molecular hydrogen. It may not be a direct effect as a scavenger. Change the description.
- The method of preparing the HW: Fig. 1 shows how to make the MHW with sticks and the concentration of molecular hydrogen, showing “pre NHW” and “post MHW” which is 3-4 days later. However, in the section of Hydrogen concentration and pH of MHW in the Methods, “a 2-week supply of MHW and DGW was prepared every 3 to 4 days by bubbling 0.03 MPa of hydrogen gas into drinking water at 25℃ for 30 min. The MHW before and after the addition of hydrogen gas was standardized as 0% and 100%, respectively. Subsequently, the amounts of dissolved MH in the stored MHW and freshly prepared MHW and DGW were calculated.” It’s confusing. Is the bubbling hydrogen gas is just for standardization? Which way was used for animals? The concentration of hydrogen gas in the MHW used in the animal experiments should be directly measured by gas chromatography.
Author Response
Dear the reviewer #1,
We appreciate your important comments for the manuscript, and have revised it as much as possible, according to each the comment.
We have revised the added parts with red letters, or the deleted ones with red double lines.
In the uploaded pdf file, could you please confirm the response to each the comment requiring the revision ?
Best regards,
Haruchika Masuda

Reviewer 2 Report
Reviewer’s comments and suggestions
In the current research, the author studied discusses the useful properties of molecular hydrogen water (MHW) on Cardio-vascular Function in Diet-Induced Obesity (DIO) mice. The study result suggested that MHW intake for 2 weeks decreased heart weight in DIO mice without significant changes in glucose level.
Furthermore, MHW intake improved cardiac hypertrophy, shortened the width of cardiomyocytes, dilated the capillaries and arterioles, activated myocardial eNOS Ser-1177 phosphorylation, and restored left ventricular function in DIO mice.
MHW intake prompted the histological conversion of hypertrophy to hyperplasia in brown adipose tissue (BAT) along with amelioration of cardiovascular protective genes MHW also activated the expansion, differentiation, and mobilization of EPCs to maintain vascular homeostasis. Therefore this study recommended that drinking MHW could be a strategy against cardiovascular disorders in obesity. Overall the manuscript is written well in terms of content and language aspects.
The manuscript needs to be revised based on the below comments
- “MHW intake has been reported to alleviate glucose intolerance, dyslipidemia, and elevated serum oxidized low-density lipoprotein levels [9,15-18]”. It would be better if the author can write up the mechanism involved in the treatment.
- The ethical number is required in the manuscript in the material and method section
- How many weeks and study design should be represented by an arrow or flow chart
- In section Hydrogen concentration and pH of MHW Subsequently, the amounts of dissolved MH in the stored MHW and freshly prepared MHW and DGW were calculated. Please explain the method
- Figure 3 and where required “The figure should be explained well I mean the adipose tissues were marked with name”
- The Stain should be mention in the legend part of figure 3
- Please use the same font for TNF and other cytokines in the paper. Representation should be similar
- “MHW intake had no effect on the BW or blood sugar of diabetic db/db mice with genetically engineered leptin receptor deficiency for 14 days; however, both parameters had significantly decreased at 3 months and 8 weeks, respectively[19]”. The sentence is contradictory, please check the meaning
- “EPC bioactivity in MetS, moderate exercise not only increases the number of EPCs but also accelerates reendothelialization ability, vasodilative NO production, and, inversely, ROS decline [48,50]”. It is not important to highlight exercise in this paper, as the authors did not do any type of this activity in their study design
- “recently reported that hydrogen-rich water improved fatty liver in a high-fat diet-induced model of nonalcoholic liver disease” what was the basis of the study proving the beneficial effect
- Please check reference number 41
Author Response
Dear the reviewer #2,
We appreciate your important comments for the manuscript, and have revised it as much as possible, according to each the comment.
We have revised the added parts with red letters, or the deleted ones with red double lines.
In the uploaded pdf file, could you please confirm the response to each the comment requiring the revision ?
Best regards,
Haruchika Masuda

Reviewer 3 Report
Comments to the authors:
-given the antioxidant effects of MH and the link between ROS, inflammation and NOS expression, could the authors show in AT and myocardium the oxidative stress level (staining for 4HNE), macrophages influx (F4/80 staining) and cytokine release (il1b tnfa by ELISA) in response to MH?
Author Response
Dear the reviewer #3,
We appreciate your important comments for the manuscript, and have revised it as much as possible, according to each the comment.
We have revised the added parts with red letters, or the deleted ones with red double lines.
In the uploaded pdf file, could you please confirm the response to each the comment requiring the revision ?
Best regards,
Haruchika Masuda
